# Short-Term Aerobic Exercise Did Not Change Telomere Length While It Reduced Testosterone Levels and Obesity Indexes in PCOS: A Randomized Controlled Clinical Trial Study

**DOI:** 10.3390/ijerph182111274

**Published:** 2021-10-27

**Authors:** Victor Barbosa Ribeiro, Daiana Cristina Chielli Pedroso, Gislaine Satyko Kogure, Iris Palma Lopes, Barbara Aparecida Santana, Hugo Celso Dutra de Souza, Rui Alberto Ferriani, Rodrigo Tocantins Calado, Cristiana Libardi Miranda Furtado, Rosana Maria dos Reis

**Affiliations:** 1Department of Gynecology and Obstetrics, Ribeirao Preto Medical School, University of São Paulo, Ribeirão Preto, São Paulo 14048-900, Brazil; victorbarbosa@ifsp.edu.br (V.B.R.); daia.pedroso@hotmail.com (D.C.C.P.); gisatyko@gmail.com (G.S.K.); irispalma20@yahoo.com.br (I.P.L.); raferria@fmrp.usp.br (R.A.F.); 2Federal Institute of São Paulo, Jacareí, São Paulo 12322-030, Brazil; 3Department of Internal Medicine, Ribeirão Preto Medical School, University of São Paulo, Ribeirão Preto, São Paulo 14048-900, Brazil; barbarasantana@hotmail.com (B.A.S.); rtcalado@fmrp.usp.br (R.T.C.); 4Department of Biomechanics, Medicine and Rehabilitation of the Locomotor Apparatus, Ribeirao Preto Medical School, University of São Paulo, Ribeirão Preto, São Paulo 14048-900, Brazil; hugocds@fmrp.usp.br; 5Drug Research and Development Center, Postgraduate Program in Translational Medicine, Federal University of Ceará, Fortaleza, Ceará 60430-275, Brazil

**Keywords:** polycystic ovary syndrome, aerobic physical training, telomere length, body composition, hyperandrogenism

## Abstract

Metabolic and hormonal outcomes of polycystic ovary syndrome (PCOS) have implications on telomere biology and physical activity may prevent telomere erosion. We sought to observe the effects of continuous (CAT) and intermittent (IAT) aerobic training on telomere length, inflammatory biomarkers, and its correlation with metabolic, hormonal, and anthropometric parameters of PCOS. This randomized controlled clinical trial study included 87 PCOS randomly stratified according to body mass index (BMI) in CAT (*n* = 28), IAT (*n* = 29) and non-training control group (CG, *n* = 30). The exercises were carried out on a treadmill, three times per week for 16 weeks. The participants’ anthropometric characteristics and biochemical and hormonal concentrations were measured before and after aerobic training or observation period, as the telomere length that was evaluated using quantitative real-time PCR. Four months of aerobic exercises (CAT or IAT) did not alter telomere length and inflammatory biomarkers in PCOS women. Obesity index as BMI and waist circumference (WC), and inflammatory biomarkers negatively affect telomeres. The hyper-andro-genism measured by testosterone levels was reduced after both exercises (CAT, *p* ≤ 0.001; IAT, *p* = 0.019). In particular, the CAT reduced WC (*p* = 0.045), hip circumference (*p* = 0.032), serum cholesterol (*p* ≤ 0.001), and low-density lipoprotein (*p* = 0.030). Whereas, the IAT decreased WC (*p* = 0.014), waist-to-hip ratio (*p* = 0.012), free androgen index (FAI) (*p* = 0.037). WC (*p* = 0.049) and body fat (*p* = 0.015) increased in the non-training group while total cholesterol was reduced (*p* = 0.010). Booth exercises reduced obesity indices and hyperandrogenism on PCOS women without changes in telomere length or inflammatory biomarkers.

## 1. Introduction

Polycystic ovary syndrome (PCOS) is a multifactorial heterogeneous endocrine disorder where the main characteristic behind this syndrome is chronic anovulation due to hyperandrogenism, a striking feature in this disease. However, the PCOS clinical expression varies and may include oligo-ovulation or anovulation and/or clinical or biochemical hyperandrogenism and evidence of polycystic ovaries [1]. Infertility and metabolic complications, such as dyslipidemia, hypertension, abnormal glucose metabolism, insulin resistance (IR), and obesity, are often present in PCOS [2], which increased the risk of developing cardiovascular disease (CVD) and type 2 diabetes mellitus (T2DM) [3,4]. This variability of phenotypes associated with PCOS depends on ethnicity and directly interferes with the prevalence of this syndrome, which affects between 5 and 16% of women of reproductive age [2].

Despite the genetic alterations related to PCOS [5], a strong environmental contribution is related to the development of the syndrome or even the worsening of the clinical conditions. The management of obesity with diet [6] or physical activity [7,8] has been suggested as a first-line treatment to improve related symptoms and infertility, a matter of concern in PCOS treatment [9]. This suggests an epigenetic component related to the pathogenesis of PCOS that affects gene expression, genomic stability, and telomere attrition [10]. Progressive telomere shortening is associated with loss of cellular proliferative capacity and premature reproductive aging, leading to chronic anovulation and infertility [11]. Several factors such as oxidative stress, inflammation, mitochondrial dysfunction, and hormonal alterations, as observed in PCOS, may accelerate telomere erosion [12]. On the other hand, increased levels of androgens in PCOS may be a protective factor improving telomerase activity [13] thereby not changing [7,14] or increasing telomere repeats [15]. These conflicting results were recently reported and are being continuously investigated [16].

It is well-known that regular practice of physical activity can improve metabolic complications and hyperandrogenism in women with PCOS, with implications in chronic anovulation and ultimately restoring fertility. Some studies have proposed that physical training could protect progressive shortening of telomeres, preventing premature aging [7,17,18]. Telomere shortening is associated with sedentarism, obesity, cardiometabolic risk factors, and oxidative stress, which leads to the development of many human diseases, in addition to a shorter life expectancy [19,20]. The intensity and interval training may have different effects on telomere biology. The aerobic physical exercise of moderate to high intensity improved metabolic and reproductive outcomes of PCOS women, reducing chronic anovulation, cardiometabolic risk, IR, and obesity-related indexes [21]. Larocca et al. (2010) [17] showed that the telomere length is more preserved in the physically active elderly compared to the inactive ones, and a positive correlation between telomeres and aerobic capacity was observed.

Previously we reported that progressive resistance training (PRT) [7,8] had positive effects on hormonal and physical characteristics of women with PCOS, with no effects on telomere length specifically related to PCOS. However, the type of physical exercise and the intensity have different effects on metabolic rate, hormonal levels, body composition, and reproductive health in women with PCOS [8,22,23] that could interfere in telomere biology. The effects of supervised aerobic physical exercise on telomere length and its implication on inflammatory biomarkers, metabolic disturbance, and reproductive outcomes of PCOS were not investigated. Considering the importance of the practice of physical exercise in women with PCOS, we now investigate the effects of two aerobic physical training protocols, continuous (CAT) and intermittent (IAT), on telomere length and its correlation with metabolic, hormonal, and anthropometric parameters in women with PCOS.

## 2. Participants, Materials and Methods

### 2.1. Study Design and Ethics Statement

This randomized, controlled, three-arm parallel-group study was approved by the Institutional Review Board of the University Hospital (UH), Ribeirao Preto Medical School-University of São Paulo (FMRP-USP) (Protocol number nº 9640/2014, and all participants gave written informed consent. The authors confirm that all ongoing and related trials for this intervention were registered in the Brazilian Clinical Trials Registry (ReBec; RBR-78qtwy) and after the study was also registered in the International Controlled Randomized Controlled Trial Registry (ISRCTN10416750).

### 2.2. Participants

To determine the clinical baseline for phenotypic disease, the volunteers were divided into two groups according to their BMI (<30 and ≥30 kg/m^2^) and then stratified into three subgroups within the two groups. The allocation group was placed inside opaque, sealed envelopes, grouped in blocks of 15 and consecutively picked depending on the BMI of the participant at the time of study inclusion. After the run-in period, 110 volunteers were randomly assigned in a 1:1:1 fashion to one of three groups (continuous aerobic training (CAT, *n* = 28), intermittent aerobic training (IAT, *n* = 29), and control group (CG), without training (*n* = 30). Random allocation was conducted by the principal investigator and participants were enrolled and assigned to the intervention groups by research assistants. The intervention groups CAT and IAT trained for 16 weeks. CG group were asked to maintain their usual daily physical activity profile. In addition, all volunteers were instructed to maintain their daily diets during the intervention.

PCOS women, aged between 18 and 39 years, who did not practice regular physical exercise (at least three times per week) were included in this study. The participants were not using any pharmacological intervention for PCOS treatment and had no dietary energy restrictions. The subjects were recruited at the Gynecological Endocrinology Outpatient Clinic of the Human Reproduction Service of the Gynecology and Obstetrics Department of Ribeirao Preto Medical School, University of São Paulo. Exclusion criteria were the presence of systemic diseases, use of drugs that interfere in the hypothalamic-pituitary-ovarian axis, congenital adrenal hyperplasia, diabetes, smoking, pregnancy, thyroid diseases, hyperprolactinemia, musculoskeletal disorders, or Cushing’s disease. PCOS diagnosis was based on the Rotterdam consensus, established on at least 2 of the following 3 features: chronic anovulation, hyperandrogenism (clinical or biochemical), and polycystic ovaries on ultrasound [24]. The presence of polycystic ovaries was determined by transvaginal pelvic ultrasonography with a Voluson E8 Expert machine (GE Healthcare, Zipf, Austria).

### 2.3. Clinical and Biochemical Measurements

Clinical characteristics as age, diastolic blood pressure, systolic blood pressure and heart rate were evaluated. The concentrations of total testosterone, androstenedione Follicle-stimulating hormone (FSH), luteinizing hormone (LH), thyroid-stimulating hormone (TSH), sex steroid hormone-binding globulin (SHBG), fasting insulin, and 17-hydroxyprogesterone (17-OHP), prolactin, estradiol, homocysteine, and c-reactive protein (CRP) were determined using a chemiluminescent method (Immulite 1000; Immunoassay System; Siemens^®^, Santa Ana, CA, USA). Fasting blood glucose was determined using the oxidase method (CMD 800X1 / CMD 800iX1, Wiener Lab, São Paulo, Brazil). Fasting High-density lipoprotein (HDL), total cholesterol, and triglycerides were evaluated using the enzymatic method (CMD 800X1 / CMD 800iX1, Wiener Lab, São Paulo, Brazil). Low-density lipoprotein (LDL) was measured using the Fried Ewald formula: LDL cholesterol = total cholesterol—(HDL cholesterol + triglycerides/5). The Free Androgen Index (FAI) was obtained from the following formula: (total testosterone (nmol/L)/SHBG (nmol/L) × 100). The homeostatic model assessment of insulin resistance (HOMA-IR) was evaluated using the formula: (fasting blood glucose in mg/dL 0.05551) fasting insulin μUI/mL/22.5.

### 2.4. Anthropometry and DXA

Body weight and height were accessed, and body mass index (BMI) was subsequently determined from these measurements. Waist circumference (WC) was measured at the midpoint between the lateral iliac crest and the lowest rib margin at the end of normal expiration. The hip circumference (HC) was measured in the region where the buttocks are largest. The waist-to-hip ratio (WHR) was obtained by dividing WC (cm) by HC (cm). All anthropometric indexes were assessed according to the procedures described by the “International Standards for Anthropometric Assessment” [25]. Body composition was measured using dual-energy X-ray absorptiometry (DXA) using the Hologic Discovery Wi, QDR series (Hologic, Inc., Waltham, MA, USA). The regions of interest (ROIs) for assessment of the total body fat (BF) (fat mass (g) plus lean mass including bone mineral content (g)) and the percentage fat (%) (fat mass/total mass × 100) were evaluated and the android and gynoid fat distributions were calculated.

### 2.5. Aerobic Physical Training Protocols

The aerobic physical training protocols were performed according to the regulations of the American College of Sports Medicine (ACSM) [26]. The training protocols were recently published by our research group [23] and are present in Appendix A. Two protocols were applied: continuous aerobic training (CAT) and intermittent aerobic training (IAT). All exercises were performed under personal supervision of a physical education professional and the measurements were carried out at baseline and after 16 weeks of training or the observational period for the control group (CG). The exercises were carried out on a treadmill (Embreex 570-L and Embreex 570-Pro, SC, Brazil), three times per week (wk) for 16 weeks, lasting equally and progressively from 30 min in the first wk, to 50 min in the last wk. The target intensity training areas followed the ACSM recommendations. Light (50–64% HRmax), moderate (64–77% HRmax) and vigorous (77–94% HRmax) intensities were considered to calculate the progression of the protocols to be applied in the context of the clinical profile of participants. To calculate the intensity of training, the HRmax formula (220-age) was used [26]. The protocols were equalized by volume at each progression. The equivalent result for each week of each protocol was added, and the total volume of IAT was similar to that of CAT. In both CAT and IAT, volunteers underwent training three times per week on nonconsecutive days.

For both protocols, five minute warmups and five minute cool downs, between 50% and 60% of the MHR, were included. The exercises were performed at the Department of Gynecology and Obstetrics and Cardiovascular Physiology and Physiotherapy Laboratory at FMRP-USP. Adherence was determined from the supervised participation of the exercise session and the conclusion of the aerobic exercise training protocols. The participants were monitored daily and in case of absence, the session was remarked for another day of the same week. The non-adherence criterion was considered to fail to participate in at least 20% of the proposed protocol training sessions. The amount of exercise/physical activity that the CG performed during the study was self-reported using recall diaries. In addition, all participants were instructed to maintain their daily diets during the intervention or observational period.

### 2.6. Telomere Length Measurement

Genomic DNA was isolated from peripheral blood leukocytes of all participants before and after the training protocols or the observation period using MasterPure Complete DNA and RNA Purification Kit (Epicentre, Illumina Company, San Diego, CA, USA), according to the manufacturer’s instructions. DNA integrity was accessed by agarose gel stained with GelRead (Unisciences, Miami Lakes, FL, USA), and the concentration was determined using the Nanodrop 2000c spectrophotometer (Thermo Fisher Scientific, Waltham, MA, USA). Telomere length was measured by the quantitative polymerase chain reaction, as described previously [27,28] using the following primer sequences for the telomere: Tel-Fw, 5′-CGGTTTGTTTGGGTTTGGGTTTGGGTTTGGGTTTGGGTT-3′ and Tel-Rv, 5′-GGCTTGCCTTACCCTTACCCTTACCCTTACCCTTACCCT-3′; and for the single gene S-Fw, 5′-CAGCAAGTGGGAAGGTGTAATCC-3′ and S-Rv, 5′-CCCATTCTATCATCAACGGGTACAA-3′.

Telomere length was determined by calculating the telomere to single-copy gene ratio using ΔCt (Ct(telomere)/Ct(single gene)). The telomere length was expressed as the relative T/S (Telomere/Single gene) ratio, normalized to the mean of the T/S ratio of the reference sample (2−(Δctx—ΔCtr) = 2-ΔΔCt). The reference sample was also used as the standard curve and validation sample. The samples that were analyzed in triplicate and reaction mix contained: 1.6 ng of genomic DNA, 7.2 pg of each primer (except for the reverse primer of the single gene, which will contain 12 pg) and 16 μL of the Rotor-Gene SYBR Green PCR Master Mix (Qiagen Hilden, Hilden, Germany). The PCR conditions used for the test were: 95 °C 5 min, 98 °C 7 s, 60 °C 10 s (25 cycles). The 100-well discs were handled using the QIAgility (Qiagen Hilden, Hilden, Germany) liquid handling instrument. The single gene reaction was obtained in 35 cycles, for 61 min and the PCR conditions were: 95 °C 5 min; 98 °C 7 s, 58 °C 10 s (35 cycles).

### 2.7. Statistical Analysis

The sample size was calculated based on the previously published data for telomere length in young PCOS women [29], using mean 0.80 and standard deviation ± 40, which showed that 30 participants (60 for each group) will be necessary to observe a difference of 0.3 in telomere length between the exercise groups and non-exercise group, 80% of statistical power, and a level of significance of 0.05. For other outcomes, the sample size was obtained considering a Cohen’s d effect size of 0.6, a significance level of 5%, and a test power of 80%.

The analysis of variables studied is presented as mean and standard deviation. To evaluate the effects of physical training (CAT and IAT) or observational period on clinical characteristics, anthropometric indices and telomere content, The Mann-Whitney nonparametric test was used to independently compare the distribution of variables between the PCOS and the control groups. A general linear mixed model (GLMM) analysis of repeated measures was used considering time, group, and group × time interactions, adjusted for the independent variables age, BMI, total testosterone and androstenedione. Residual analysis was performed to verify the adjustments of the statistical models. To compare the groups, orthogonal contrasts were used considering linear models of mixed effects. Additionally, an analysis of the correlation between the telomere length and quantitative PCOS variables (age, BMI, total testosterone and androstenedione) were assessed using Pearson correlation coefficients. All statistical analyses were performed using SAS^®^ 9.3 software (SAS Institute Inc., University of North Carolina, Cary, NC, USA), and *p* < 0.05 was considered significant.

## 3. Results

The flowchart of the study is illustrated in Figure 1. According to the eligibility criteria, 126 participants were recruited. Of these 126, 16 were unable to reach the inclusion criteria for PCOS after initial evaluations, thus 110 women with PCOS started the physical training protocols. Of these, 23 did not finish the protocols and 87 participants completed the study: 28 in CAT, 29 in the IAT, and 30 in the CG groups. To adhere to the protocols and complete the study, the adherence was at least 90% of the training sessions.

The physical, anthropometric and hormonal characteristics of the groups analyzed before and after the training or the observational period are presented in Table 1. The age, diastolic and systolic blood pressure were not different between the studied groups. To characterize the PCOS, prolactin (CG = 16.6 ng/mL ± 9.1; CAT =17.4 ng/mL ± 12.7; IAT = 16.8 ng/mL ± 11.7), 17-OHP (CG = 106.0 uUI/mL ± 38.0; CAT = 98.0 uUI/mL ± 47.0; IAT = 86.0 uUI/mL ± 40.0), and TSH (CG = 2.38 ng/dL ± 1.18; CAT =1.76 ng/dL ± 0.67; IAT = 2.64 ng/dL ± 1.60) were measured. At baseline, the total testosterone level was higher in the CAT group (117 ± 50 ng/dL) when compared to the CG (86 ± 37 ng/dL), *p* = 0.01. The other variables analyzed were not different at the beginning of the training protocols. Serum levels of androstenedione, SHBG, estradiol, FSH, and LH did not change after aerobic physical training protocols (CAT and IAT) and the observational period in CG. The testosterone level decreased after CAT (*p* ≤ 0.001) and IAT (*p* = 0.019) and the FAI was reduced only in the IAT group (*p* = 0.037).

After the aerobic physical exercises or no training periods, no differences were observed in the anthropometric indices BMI and weight, or the metabolic parameters such as HDL, triglycerides, fasting glycemia and insulin and HOMA-IR. The lipidic profile as total cholesterol (*p* ≤ 0.001) and LDL (*p* = 0.030) was reduced after CAT. The anthropometric evaluation showed that WC and HC were reduced in the CAT group (*p* = 0.045 and *p* = 0.032, respectively), as well as the heart hate (*p* = 0.016), and the WC and WHR decreased in the IAT group (*p* = 0.014 and *p* = 0.012, respectively). In the control group without training (CG), an increase in WC (*p* = 0.049) and body fat percentage (*p* = 0.015) was observed and a reduction in total cholesterol (*p* = 0.010). No differences were observed in other anthropometric or DXA variables evaluated.

Regarding the effects of CAT and IAT or non-training on telomere length, we observed great variability in the subjects in both training and non-training control groups (Figure 2), however in the adjusted model the telomere length did not change after CAT (*p* = 0.6912), IAT (*p* = 0.9099) or CG (*p* = 0.6028).

Data are presented as mean and standard deviation (SD). CG. control group; CA, continuous aerobic physical training group; AI, intermittent aerobic physical training group; ng/dL, nanogram per deciliter; nmol/L, nanomolar per liter; mg/dL, milligrams per deciliter; pg/mL, picogram per milliliter; uIU/mL, international microunits/milliliter; BMI. body mass index; %. fat percentage; cm. centimeters; WC. waist circumference; HC. hip circumference; WHR. waist hip ratio; DBP, diastolic blood pressure; SBP, systolic blood pressure; BPM, beats per minute; SHBG, sex hormone binding globulin; FAI, free testosterone index; E2, estradiol; LH, luteinizing hormone; FSH, follicle stimulating hormone; HDL, High Density Lipoproteins; LDL, Low Density Lipoproteins; HOMA-IR, homeostatic model assessment; T/S ratio, Telomere/single gene. In bold, *p* < 0.05 intra-group. ^a^, *p* < 0.05 intergroup. * Covariate adjusted Covariate-adjusted analyses (BMI, Age and testosterone and Androstenedione levels).

Regardless of continuous and intermittent aerobic training, four months of aerobic exercise seems not change telomere length (Figure 3A). Indeed, the inflammatory biomarkers’ homocysteine and c-reactive protein were not altered after the aerobic training, nor in the control non-training groups (*p* > 0.05). In the general linear mixed models, the confoundable variables, such as age (*p* = 0.4204), BMI (*p* = 0.2983), total testosterone (*p* = 0.7115), androstenedione (*p* = 0.8993), did not interfere in telomere length in these groups, even the independent variables group (*p* = 0.2552) and time (*p* = 0.4098). The correlation analysis showed that the anthropometric variables age (*r* = –0.1633, *p* = 0.0324), BMI (*r* = −0.1784, *p* = 0.0192) and WC (*r* = −0.1613, *p* = 0.0334) (Table 2, Figure 3B,C, respectively) had a negative correlation with telomere length, as well as the inflammatory biomarkers CRP (*r* = –0.2161, *p* = 0.0041) and homocysteine (*r* = −0.1635, *p* = 0.0311). The hyperandrogenism, including total testosterone and androstenedione, was not correlated with telomere length in women with PCOS (Table 2).

## 4. Discussion

The importance of physical training in the treatment of PCOS needs to be clarified. In the present study, we did not observe differences in telomere length after the CAT and IAT training protocols, with no changes in the inflammatory biomarkers’ homocysteine and C-reactive protein. Despite no differences in telomere length and inflammation, the CAT protocol reduced the hormonal level of total testosterone, total cholesterol, and LDL, while also improving anthropometric indexes, decreasing WC and HC. The IAT, on the other hand, showed a decrease in total testosterone and FAI, and a reduction in WC and WHR. The non-training CG’s total cholesterol was reduced; however, the WC and body fat percentage were increased after the non-intervention observational period. Considering the evaluation of structured aerobic physical training protocols in PCOS women, most studies are transversal and unsupervised, and the effects of physical training on telomere length may not be representative [19].

To our knowledge, few studies investigate changes in telomere length in PCOS, and the results are controversial leading to ambiguous conclusions [16], probably due to the heterogeneity of this syndrome. The first study was carried out by Li et al. (2014) and the authors observed leukocytes telomere shortening in PCOS women [29]. Then, we reported no alterations in leukocyte’s telomere length from PCOS, however, a negative correlation of inflammatory biomarkers with telomere length was observed [14]. In agreement, Wei et al. (2017) did not find changes in telomere length in leukocytes of PCOS women [30]. Contradictorily, an increase in leukocytes’ telomeres from women with PCOS was reported by Wang et al. (2017). The authors also reported a reduction in the TERRA (Telomeric repeat-containing RNA) gene expression that was negatively correlated with testosterone, whereas telomere length had a positive correlation with testosterone [15].

We previously reported that physical resistance training (PRT) improved hyperandrogenism, reproductive function, and body composition in women with PCOS, with no changes in BMI or metabolic parameters of PCOS [8]. The PRT was performed three times per week for four weeks in PCOS and non-PCOS women. We also observed that PRT reduced telomere length and increased the homocysteine level in all participants, irrespective of them having PCOS or not. Considering aging and senescence, the PRT should be practiced with caution. In addition, a positive correlation between androstenedione and telomere length was observed in PCOS [7], suggesting that hyperandrogenism may be an important protective factor for telomere erosion, irrespective of the metabolic disturbance that contributes to telomere shortening in PCOS [13]. Short term PRT seems to change telomere content, however, the same period of continuous or intermittent aerobic training did not affect telomere biology.

Loprinzi and Sng (2016) evaluated the effect of nine types of unsupervised exercise on telomere length and suggested that running was the most effective exercise with positive impacts on telomere biology [31]. In young and elderly populations, the endurance exercise practice increases maximum aerobic capacity (VO2max) that consequently increases telomere length [17,32]. Unfortunately, we did not measure the VO2max in our groups. Tucker (2017) [33] evaluated the metabolic equivalent (MET), the ratio of the work metabolic rate to the resting metabolic rate using frequency, intensity and duration, and observed that a higher level of physical activity meant longer telomeres were reported in men and women. Differently, shortening of telomeres in endurance athletes after exposure to acute exercise was observed, due to oxidative damage in DNA [34].

Metabolic alterations such as IR, dyslipidemia and low HDL-cholesterol, obesity, body fat and most recently inflammation, increases the risk of developing DMT2 and CVD morbidities/mortality [4]. Non-pharmacological treatment, as physical exercise, can improve the cardiometabolic profile in women with PCOS without weight loss. High intensity interval aerobic training reduced insulin resistance and body composition in women with PCOS, whereas the resistance exercise improved the body composition that was independent of weight loss [35]. We also investigated the effects of aerobic exercise on metabolic, hormonal and anthropometric parameters which also may interfere in telomere biology. Both CAT and IAT showed benefits to women with PCOS, decreasing total testosterone level and WC in PCOS women. Specifically, the progressive moderate intensity CAT exercise reduced HC, heart rate, total cholesterol and LDL level, while IAT with progressive moderate/vigorous intensity reduced WHR and FAI. Mario et al. (2017) [36] reported that habitual physical activity improves anthropo-metric parameters and hyperandrogenism in women with PCOS. According to non-supervised data, intense physical activity reduces metabolic syndrome, irrespective of age, BMI, or total energy expenditure [22] and moderate intensity aerobic exercise reduces cardiometabolic risks factors in PCOS [37]. On the other hand, the non-training CG increased WC, body fat (%) obesity indexes that are related to IR, hyperinsulinemia and hyperandrogenism in PCOS [38]. The reduction in total and LDL-cholesterol in the CG may be due to some clinical characteristics that were not evaluated, or differences in dietary habits since we did not control this variable.

As a molecular marker of aging, progressive telomere shortening with each cell division is a common biological process, leading to chromosomal instability and cell senescence or apoptosis. Age-related diseases and some metabolic alterations might contribute to an accelerated shortening of telomeres [39] while other conditions, such as increased estrogen levels [13], and physical activity prevent progressive telomeric loss [40]. As reported in this study, Mason et al. (2013) did not observe changes in telomere biology after aerobic physical exercise, and maybe the time of exposition was relatively shorter to observe changes in telomere length [41]. Since telomere changes are progressive and a relatively slow process (~52 base pair/year) [42], 16 weeks of intervention maybe not be sufficient to observe the effect of the aerobic intervention on telomere biology. Sanberoth et al. (2015) [43] suggested that at least 10 years of regular physical activity will be necessary for improvements on telomeres, which may explain our results, since four months of CAT or IAT might be insufficient to observe differences in telomere biology, and an increase in exposure time might provide additional information. Despite no differences in telomeres, in the non-obese population, physical exercise may prevent telomere shortening [17,32].

In the adjusted model of our data, none of the confoundable variables seemed to interfere in telomere length in PCOS. However, a negative correlation was observed between telomere length and age, as expected, and the obesity indices BMI and WC and the inflammatory biomarkers CRP and homocysteine. A cross-sectional meta-analysis of 87 observational studies reported that telomere length is negatively correlated with BMI, since higher BMI is related to reduced telomeres, and this association appeared to be stronger in young individuals [44]. This negative correlation may be due to the increased inflammation and oxidative stress, a common characteristic of PCOS that is positively correlated with BMI [45]. Nevertheless, the inflammatory biomarkers did not change after aerobic exercises, but a negative correlation with telomere length was observed in PCOS as previously reported [14]. Increased WC is related to reduced telomeres in obese women and physical activity is an important strategy for obesity treatment, by reducing anthropometric indices and improving body composition [40,46]. Since booth exercises reduced WC and hyperandrogenism in our BMI randomized study, the differences in telomere length might not be observed in a short time of training. Associated with the well-controlled study, as a repetitive molecular marker, telomeres have great variability.

To observe the effect of the supervised exercise we did not control the diet of the participants or evaluate the levels of habitual physical activity related to work and leisure. Physical activity is also related to gastrointestinal GUT microbiome changes and may improve metabolic disorders related to PCOS [47], by promoting an anti-inflammatory environment [48]. CRP and homocysteine inflammatory biomarkers did not change after 16 weeks of CAT or IAT, however, microbiome evaluation is an interesting topic for future studies in PCOS. Despite the important findings of this manuscript, one of the limitations of our clinical trial is that we used platform chemiluminescence for the measurement of steroid sex hormones, which may not be as sensitive as mass spectrometry for detecting androgen levels in women. It is important to consider that intensity of the protocols was defined in terms of HRmax (%). Although the literature shows which there is a linear relationship between VO2max and HRmax at submaximal exercise intensities, the VO2max (%) or metabolic equivalent (METS) are considerate the gold standard to determine exercise intensity [22]. It may have underestimated the intensity. However, in this controlled clinical trial study, the effect of aerobic physical exercises evaluated was supervised by a physical education professional without dietary restrictions, which strengthens our results.

## 5. Conclusions

Short-term aerobic physical intervention, continuous and intermittent training, did not promote changes in telomere length and inflammatory parameters, while it reduces testosterone levels and improved anthropometric indices in PCOS women. Nevertheless, telomeres were negatively affected by obesity indices, such as BMI and WC, and inflammatory biomarkers CRP and homocysteine. Additionally, after the CAT, the WC and HC were reduced, as well as total cholesterol, LDL, and total testosterone, whereas the IAT only showed a reduction in WC and WHR, and a decrease in total testosterone level and FAI. The non-exercise practice in the CG increased WC and body fat. Booth aerobic physical training protocols should be considered as an effective strategy in the treatment of metabolic disorders and hyperandrogenism in PCOS women, but the implications on telomere biology should be investigated over a longer period than the one in this study, with the use of these aerobic training protocols.

## Figures and Tables

**Figure 1 ijerph-18-11274-f001:**
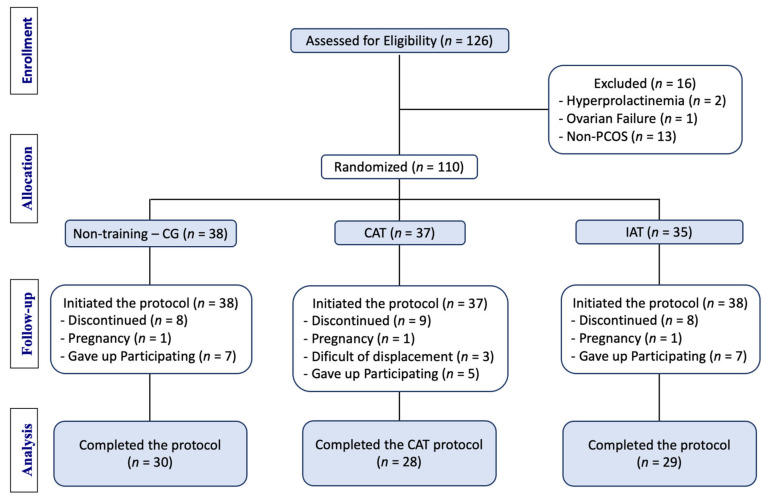
Flowchart of the study.

**Figure 2 ijerph-18-11274-f002:**
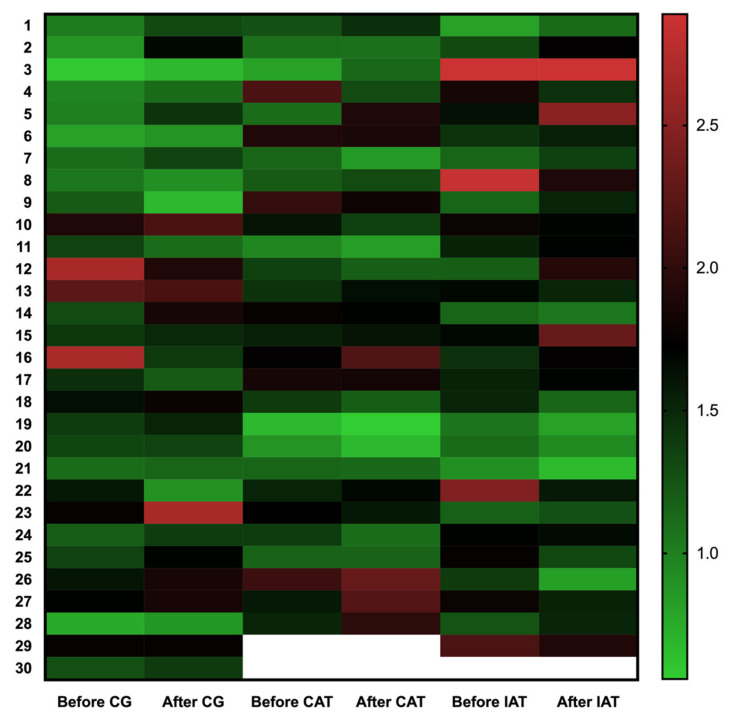
Heatmap of telomere length (T/S ratio) variation before and after 16 weeks of continuous and intermittent aerobic physical training or observation period.

**Figure 3 ijerph-18-11274-f003:**
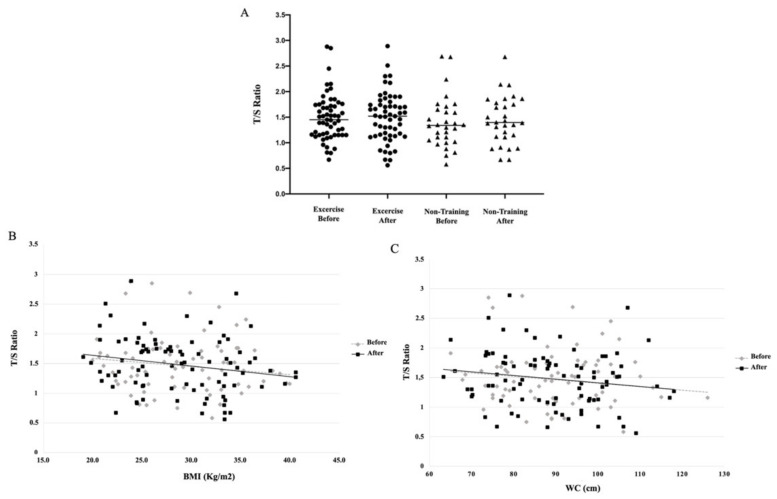
Telomere length (T/S ratio) before and after 16 weeks of aerobic physical training or observation period (*p* > 0.05) (**A**). Spearman correlation between telomere length vs. BMI (*p* = 0.0192) (**B**) and WC (**C**) (*p* = 0.0334). CG, Control group; CAT, continuous aerobic training; IAT, intermittent.

**Table 1 ijerph-18-11274-t001:** Clinical, anthropometric, hormonal and metabolic parameters of women with PCOS before and after the aerobic physical training protocols or observation period.

	CG (*n* = 30)		CAT (*n* = 28)		IAT (*n* = 29)	
Variables	Before Mean (SD)	After Mean (SD)	Before Mean (SD)	After Mean (SD)	Before Mean (SD)	After Mean (SD)
Age (years)	28.50 (5.76)	–	29.14 (5.26)	–	28.97 (4.32)	–
Height (m)	1.61 (0.07)	–	1.62 (0.06)	–	1.64 (0.07)	–
Weight (kg)	75.37 (14.33)	76.05 (15.09)	74.4 (16.5)	73.74 (16.78)	77.36 (16.91)	77.00 (16.81)
BMI (kg/m^2^)	29.09 (5.25)	29.33 (5.43)	28.43 (5.62)	28.17 (5.67)	28.67 (4.76)	28.53 (4.82)
WC (cm)	89.52 (12.61)	**90.98 (13.14)**	88.12 (13.60)	**86.58 (13.12)**	90.54 (11.33)	**88.67 (12.43)**
HC (cm)	106.34 (10.15)	107.23 (9.75)	105.88 (9.58)	**104.55 (10.27)**	107.31 (9.50)	107.17 (10.98)
WHR (cm)	0.84 (0.08)	0.85 (0.07)	0.83 (0.08)	0.82 (0.07)	0.84 (0.06)	**0.83 (0.07)**
SBP (mm Hg)	105.07 (11.29)	108.13 (10.42)	104.36 (9.39)	100.39 (9.88)	104.69 (12.12)	103.93 (14.19)
DBP (mm Hg)	71.40 (10.11)	72.00 (8.89)	69.64 (9.85)	68.25 (9.02)	68.52 (9.38)	68.52 (11.38)
Heart Rate (bpm)	74.93 (11.17)	76.90 (10.00)	76.82 (12.44)	**71.86 (9.52)**	73.90 (9.98)	73.86 (12.34)
**Biochemical parameters**						
Testosterone (ng/dL)	86.2 (37) ^a^	99.67 (46.40)	116.7 (49.5) ^a^	**92.7 (37.8)**	107.69 (51.53)	**87.8 (54.2)**
Androstenedione (ng/dL)	87.2 (56)	78.63 (45.60)	86.64 (44.61)	82.0 (28)	77.69 (59.75)	74.52 (49.42)
SHBG (nmol/L)	50.56 (34.21)	62.26 (45.29)	54.31 (40.89)	58 (61)	47.82 (28.19)	53.49 (31.00)
FAI	7.52 (4.21)	7.90 (5.98)	11.33 (9.58)	10.3 (10)	9.87 (7.20)	**7.84 (7.72)**
E2 (pg/mL)	48.12 (23.73)	48.31 (22.07)	55.83 (42.84)	67 (69)	63.44 (55.42)	61.41 (48.73)
LH (μUI/mL)	9.25 (8.40)	6.78 (4.81)	7.79 (4.76)	8.32 (8.56)	7.77 (4.87)	7.63 (4.74)
FSH (μIU/mL)	5.63 (1.99)	5.16 (1.78)	5.66 (2.14)	5.12 (1.83)	5.21 (2.95)	5.59 (2.71)
Total Cholesterol (mg/dL)	188.27 (34.13)	**177.53 (24.44)**	184.64 (29.83)	**171.45 (28.07)**	178.86 (29.34)	174.17 (26.86)
Triglycerides (mg/dL)	111.77 (55.82)	103.17 (58.87)	151.43 (172.63)	144.35 (139.16)	98.62 (54.49)	106.83 (60.85)
HDL (mg/dL)	50.10 (13.09)	48.47 (12.66)	45.67 (9.33)	44.28 (10.29)	48.78 (10.62)	47.08 (10.27)
LDL (mg/dL)	115.73 (31.55)	**108.33 (26.80)**	111.71 (23.55)	**102.46 (23.14)**	112.31 (23.49)	106.24 (23.19)
Fasting Glycemia (mg/dL)	83.0 (7.0)	81.0 (9.0)	84.0 (12.0)	84.0 (11.0)	82.0 (11.0)	82 (11.0)
Fasting Insulin (μIU/mL)	12.83 (8.5)	12.45 (9.79)	11.31 (8.09)	11.2 (8.3)	9.52 (7.18)	10.40 (7.03)
HOMA-IR	2.64 (1.77)	2.59 (2.17)	2.45 (1.90)	2.42 (1.97)	2.02 (1.88)	2.20 (1.79)
Homocysteine (µmol/L)	7.62 (1.74)	7.89 (1.64)	8.05 (2.40)	7.66 (1.70)	7.17 (1.80)	7.54 (1.94)
C-reative protein (mg/dL)	0.51 (0.41)	0.53 (0.54)	0.36 (0.41)	0.43 (0.54)	0.36 (0.32)	0.35 (0.32)
**Body composition (DXA)**						
Body Fat (%)	40.59 (6.26)	**41.83 (4.36)**	40.25 (4.67)	**39.20 (5.63)**	41.97 (3.79)	41.80 (5.53)
Android (%)	42.33 (8.09)	43.90 (3.08)	42.25 (6.30)	40.69 (7.13)	43.37 (4.77)	43.17 (5.60)
Gynoid (%)	43.79 (6.10)	44.e93 (4.26)	43.58 (4.97)	42.41 (5.60)	45.23 (4.59)	44.22 (5.54)
Fat Mass_height^2^ (kg/m^2^)	11.80 (3.45)	12.13 (3.08)	11.43 (3.38)	11.09 (3.49)	11.80 (2.60)	11.67 (2.74)
Lean Mass_height^2^ (kg/m^2^)	16.80 (2.76)	16.53 (2.53)	16.51 (2.49)	16.64 (2.45)	16.11 (2.39)	16.32 (2.61)
**Telomere Length**						
T/S ratio *	1.40 (0.50)	1.45 (0.46)	1.43 (0.39)	1.45 (0.46)	1.53 (0.46)	1.54 (0.51)

**Table 2 ijerph-18-11274-t002:** Pearson correlation analysis between telomere length and quantitative variables in PCOS women, regardless of group.

Variables	T/S Ratio	
	*r*	*p*-Value
Age (years)	−0.1633	**0.0324**
BMI (kg/m^2^)	−0.1770	**0.0194**
WC (cm)	−0.1613	**0.0334**
Testosterone (ng/dL)	0.1027	0.1773
Androstenedione (ng/dL)	0.0090	0.9057
CRP (mg/dL)	−0.2161	**0.0041**
Homocysteine (µmol/L)	−0.1635	**0.0311**

BMI, Body mass index; WC, Waist Circumference, CRP, C-reactive Protein, in bold *p* < 0.05.

## Data Availability

Not applicable.

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
