# Peer review of "Short-Term Aerobic Exercise Did Not Change Telomere Length While It Reduced Testosterone Levels and Obesity Indexes in PCOS: A Randomized Controlled Clinical Trial Study"

_ijerph, 2021, doi:10.3390/ijerph182111274_

Round 1
Reviewer 1 Report
The paper deals with Short-term aerobic exercise did not change telomere length while it reduces testosterone levels and obesity indexes in PCOS: a randomized controlled clinical trial study. Interesting Results part and a well done paper, in general. Please see bellow my few suggestions for improving the manuscript:
As the unit of measure for volume in your research, please replace ml with mL (as Litter being the international unit of measure for volume). Please check/revise the entire manuscript in this regard, being consistent with this denotation.
Introduction. Is it menopause (in all its stages) an associated factor for PCOS development? Does this period with drastic hormonal changes influencing PCOS? Please add a short paragraph and detail (you may refer to Tit D.M., et al. Somatic-vegetative symptoms evolution in postmenopausal women treated with phytoestrogens and hormone replacement therapy. Iran. J. Public Health 2017, 46(11), 1128-1134.
Figure 1 is blurred. Please replace it with a better quality one (I suggest print screen instead of other type of saving figures).
Table 1. Please explain under the table what is the meaning of the bolded marked numerical values and of * symbol.
Figure 3. On the B and C graphics, please check and correct the numerical values to be in english style (with point, not with comma)
Table 2. Please complete the head of the Table for the first column: Parameters or Quantitative variables in PCOS. For each followed parameter in the first column, please add their units of measure (i.e. Age (years)). Please also explain under the table what is the meaning of the bolded values in the last column.
L346-347. Please reference it. I suggest Gheorghe G., et al. Cardiovascular Risk and Statin Therapy Considerations in Women. Diagnostics 2020,10(7), 483. https://doi.org/10.3390/diagnostics10070483 and Vesa, C.M.; et al. Current Data Regarding the Relationship between Type 2 Diabetes Mellitus and Cardiovascular Risk Factors. Diagnostics 2020, 10, 314. https://doi.org/10.3390/diagnostics10050314
Author Response
The paper deals with Short-term aerobic exercise did not change telomere length while it reduces testosterone levels and obesity indexes in PCOS: a randomized controlled clinical trial study. Interesting Results part and a well done paper, in general. Please see bellow my few suggestions for improving the manuscript:
Thank you for your considerations and the time spent evaluating our manuscript. Your suggestions really improved the quality of our work. We revised the manuscript according to the suggestions made and the main changes are highlighted in red
As the unit of measure for volume in your research, please replace ml with mL (as Litter being the international unit of measure for volume). Please check/revise the entire manuscript in this regard, being consistent with this denotation.
Thank you for this important observation. The manuscript was revised, and ml and dl were replaced with mL and dL.
Introduction. Is it menopause (in all its stages) an associated factor for PCOS development? Does this period with drastic hormonal changes influencing PCOS? Please add a short paragraph and detail (you may refer to Tit D.M., et al. Somatic-vegetative symptoms evolution in postmenopausal women treated with phytoestrogens and hormone replacement therapy. Iran. J. Public Health 2017, 46(11), 1128-1134.
PCOS and menopause are note related diseases. PCOS anovulatory condition is mainly due to hyperandrogenism and metabolic alterations, while menopause or ovarian failure is due to elevated FSH and LH levels, and hypoestrogenism. Also, post-menopause PCOS women have an increased risk of cardiovascular disease and disturbance in glucose metabolism due to insulin resistance in reproductive life. There is no scientific evidence of specific symptoms arising from postmenopausal hormonal disturbances in women with PCOS.
Figure 1 is blurred. Please replace it with a better quality one (I suggest print screen instead of other type of saving figures).
We improved the quality of figure 1
Table 1. Please explain under the table what is the meaning of the bolded marked numerical values and of * symbol.
The *symbol was just for telomere which was adjusted by confoundable variables. We have changed the table.
*Covariate adjusted Covariate-adjusted analyses (BMI, Age and testosterone and Androstenedione levels)
Figure 3. On the B and C graphics, please check and correct the numerical values to be in english style (with point, not with comma)
We corrected the numerical values.
Table 2. Please complete the head of the Table for the first column: Parameters or Quantitative variables in PCOS. For each followed parameter in the first column, please add their units of measure (i.e. Age (years)). Please also explain under the table what is the meaning of the bolded values in the last column.
Sorry for this mistake. The label “Quantitative Variables” was added to the column and the units of measure.
L346-347. Please reference it. I suggest Gheorghe G., et al. Cardiovascular Risk and Statin Therapy Considerations in Women. Diagnostics 2020,10(7), 483. https://doi.org/10.3390/diagnostics10070483 and Vesa, C.M.; et al. Current Data Regarding the Relationship between Type 2 Diabetes Mellitus and Cardiovascular Risk Factors. Diagnostics2020, 10, 314. https://doi.org/10.3390/diagnostics10050314
Thank you for this suggestion. Since the first-line treatment suggested for PCOS is dietary control and regular practice of exercise, we focused on the non-pharmacological intervention to treat PCOS related symptoms. However, if you believe that is necessary, we can further add something regarding pharmacological treatment. We have added the second reference that matches with our main goal. Lines 47-48; 367-369
Reviewer 2 Report
In the present paper, Victor Barbosa Ribeiro and coworkers investigated in a randomized, controlled clinical study the effects of continuous (CAT) and intermittent (IAT) aerobic training on telomere length, inflammatory biomarkers, and its correlation with metabolic, hormonal and anthropometric parameters of polycystic ovary syndrome (PCOS). The authors showed that both exercises reduced obesity indices and hyperandrogenism on PCOS women without changes in telomere length or inflammatory biomarkers.
Overall, I think that the present paper is intriguing, timely and it could be of interest to the readers of “International Journal of Environmental Research and Public Health” and researchers, in general.
I would like to make some suggestions on how to make the paper stronger.
Research has clearly shown that healthy eating habits and regular physical activity helps to manage PCOS. The authors, if possible, could incorporate in tables the dietary pattern of the patients included in the present study (i.e. Mediterranean-style diet, Plants-based diet, Nordic dietary pattern, etc.); in this way, I feel that the readers can better understand the intriguing results obtained in the present clinical study and their possible application to current practice.
Current evidence has shown the involvement of the gut microbiome in PCOS. Does the authors plan to assess/consider microbioma composition in their patients? Please make a comment in the discussion section of revised manuscript on a hot topic of current research.
There is recent evidence that nutraceuticals as Inositols could have a positive therapeutic role in PCOS. Please discuss this topic and for your convenience you could consider the following reference (Genazzani AD; Reprod Biomed Online 2016 Dec;33(6):770-780) in your paper.
The authors could add in Graphical form the molecular pathways explored in these experiments; in this way, I feel that the readers can better understand the pathophysiological cross-talk studied in the present paper and the possible therapeutic potential of nutraceutics and/or antioxidants/antinflammatory in combination with healthy diet and regular aerobic training in the management of patients with PCOS and metabolic syndrome/obesity.
Author Response
In the present paper, Victor Barbosa Ribeiro and coworkers investigated in a randomized, controlled clinical study the effects of continuous (CAT) and intermittent (IAT) aerobic training on telomere length, inflammatory biomarkers, and its correlation with metabolic, hormonal and anthropometric parameters of polycystic ovary syndrome (PCOS). The authors showed that both exercises reduced obesity indices and hyperandrogenism on PCOS women without changes in telomere length or inflammatory biomarkers.
Overall, I think that the present paper is intriguing, timely and it could be of interest to the readers of “International Journal of Environmental Research and Public Health” and researchers, in general.
We are very thankful for your review and great considerations that improved the quality of our work. We revised the manuscript according to the suggestions made and the main changes are highlighted in red
I would like to make some suggestions on how to make the paper stronger.
Research has clearly shown that healthy eating habits and regular physical activity helps to manage PCOS. The authors, if possible, could incorporate in tables the dietary pattern of the patients included in the present study (i.e. Mediterranean-style diet, Plants-based diet, Nordic dietary pattern, etc.); in this way, I feel that the readers can better understand the intriguing results obtained in the present clinical study and their possible application to current practice.
Thank you for this important observation. The management of obesity with diet and/or physical activity in women with PCOS is an important treatment to improve the symptoms of IR and hyperandrogenism, as well as infertility. However, we sought to observe the effect of isolated aerobic physical activity on PCOS related comorbidities and body composition, without structured dietary energy restrictions, so we did not control diet in these participants. This information was added in the discussion (Lines 420-422)
Current evidence has shown the involvement of the gut microbiome in PCOS. Does the authors plan to assess/consider microbioma composition in their patients? Please make a comment in the discussion section of revised manuscript on a hot topic of current research.
This is a recent and important approach to non-therapeutic interventions related to PCOS, especially diet and physical activity, which may change the GUT microbiome diversity. However, we do not have the sample to perform this type of examination, which will certainly be the goal of our future research. We have added a comment about this issue in the discussion. (Lines 423-426)
There is recent evidence that nutraceuticals as Inositols could have a positive therapeutic role in PCOS. Please discuss this topic and for your convenience you could consider the following reference (Genazzani AD; Reprod Biomed Online 2016 Dec;33(6):770-780) in your paper.
The non-pharmacological management of obesity with diet and/or physical activity in women with PCOS is first-line treatment that improves the symptoms of IR and hyperandrogenism, as well as infertility, without any pharmacological intervention. Despite the importance of these studies, as our main goal was to observe the effect of isolated aerobic physical activity on PCOS related comorbidities and body composition, pharmacological intervention (insulin sensitizers or inositol’s) was not mentioned in this manuscript. However, if you believe that is necessary, we can further add something regarding pharmacological treatment.
The authors could add in Graphical form the molecular pathways explored in these experiments; in this way, I feel that the readers can better understand the pathophysiological cross-talk studied in the present paper and the possible therapeutic potential of nutraceutics and/or antioxidants/antinflammatory in combination with healthy diet and regular aerobic training in the management of patients with PCOS and metabolic syndrome/obesity.
Thank you for your suggestion. We have made a Graphical Summary (Figure 4) of the main results which we believe facilitated the readers' understanding.
Reviewer 3 Report
I read with great interest the manuscript, which falls within the aim of this Journal. In my honest opinion, the topic is interesting enough to attract the readers’ attention. Nevertheless, authors should clarify some points and improve the discussion, as suggested below.
Authors should consider the following recommendations:
- Manuscript should be further revised in order to correct some typos and improve style.
- Accumulating evidence suggests that one of the most important mechanisms of PCOS pathogenesis is the insulin-resistance. For this reason, the use of insulin-sensitizers, such as inositol isoforms, gained increasing attention due to their safety profile and effectiveness. Authors may better discuss this point, taking to account these recent articles: PMID: 26927948; PMID: 27579037.
Author Response
I read with great interest the manuscript, which falls within the aim of this Journal. In my honest opinion, the topic is interesting enough to attract the readers’ attention. Nevertheless, authors should clarify some points and improve the discussion, as suggested below.
Authors should consider the following recommendations:
Manuscript should be further revised in order to correct some typos and improve style.
Thank you for your considerations and the time spent evaluating our manuscript. Your suggestions really improved the quality of our work. The manuscript was revised to correct typos and improve English style. We revised the manuscript according to the suggestions made and the main changes are highlighted in red.
Accumulating evidence suggests that one of the most important mechanisms of PCOS pathogenesis is the insulin-resistance. For this reason, the use of insulin-sensitizers, such as inositol isoforms, gained increasing attention due to their safety profile and effectiveness.Authors may better discuss this point, taking to account these recent articles: PMID: 26927948; PMID: 27579037.
Thank you for this suggestion. The non-pharmacological management of obesity with diet and/or physical activity in women with PCOS is first-line treatment that improves the symptoms of IR and hyperandrogenism, as well as infertility, without any pharmacological intervention. Despite the importance of these studies, as our main goal was to observe the effect of isolated aerobic physical activity on PCOS related comorbidities and body composition, pharmacological intervention (insulin sensitizers or inositol’s) was not mentioned in this manuscript. However, if you believe that is necessary, we can further add something regarding pharmacological treatment.
Round 2
Reviewer 2 Report
Thank you for addressing my comments well. I have no further remarks.